# Single platelet variability governs population sensitivity and initiates intrinsic heterotypic responses

Maaike S. A. Jongen [1], Ben D. MacArthur[1,2,3], Nicola A. Englyst[1,3] & Jonathan West [1,3] ✉

Investigations into the nature of platelet functional variety and consequences for homeostasis require new methods for resolving single platelet phenotypes. Here we combine droplet microfluidics with flow cytometry for high throughput single platelet function analysis. A large-scale sensitivity continuum was shown to be a general feature of human platelets from individual donors, with hypersensitive platelets coordinating significant sensitivity gains in bulk platelet populations and shown to direct aggregation in droplet-confined minimal platelet systems. Sensitivity gains scaled with agonist potency (convulxin > TRAP-14>ADP) and reduced the collagen and thrombin activation threshold required for platelet population polarization into pro-aggregatory and pro-coagulant states. The heterotypic platelet response results from an intrinsic behavioural program. The method and findings invite future discoveries into the nature of hypersensitive platelets and how community effects produce population level responses in health and disease.

[1] Faculty of Medicine, University of Southampton, Southampton SO17 1BJ, UK. [2] Mathematical Sciences, University of Southampton, Southampton SO17 1BJ, UK. [3] Institute for Life Sciences, University of Southampton, Southampton SO17 1BJ, UK. ✉email: J.J.West@soton.ac.uk

Understanding cellular diversity and interactions provides the key to elucidating system behaviour. It becomes meaningful to investigate cellular diversity and identify even potentially rare phenotypes when amplification mechanisms exist in the system and when there is good reason to predict large-scale variety. Classically, cancer[1–3], immunology[4,5] and stem cells[6,7] with associated cell expansion have been the focus of the large majority of single-cell studies.

In this work we turn our attention to platelets, dispersed sentinels which patrol the vasculature to detect breaches and respond in a coordinated manner using rapid and potent paracrine signalling to collectively form a thrombus. Platelets are also inherently variable[8], originating from the fragmentation of heterotypic[9] megakaryocytes resulting in variously small subcellular compartments (60% volume CV)[10] with dissimilar contents and biochemistry[11–13] and, without a nucleus, having limited repair capabilities during ageing[14,15] before clearance. Therefore, platelet activation represents an ideal system for investigating cellular diversity and consequences for homoeostatic control. Indeed, the nature and functional consequences of platelet diversity has been a matter of enquiry for almost half a century[8,10,11]. More recently, the discovery that dual stimulation with collagen and thrombin[16,17] polarises platelets into distinct pro-coagulant and pro-aggregatory phenotypes[8,16,18–21] has renewed interest on the topic of platelet diversity. In particular, pro-coagulant platelets have been further characterised[22–27], revealing diverse functions that either represent multiple pro-coagulant subpopulations or a unified, yet versatile pro-coagulant subpopulation[21]. The bifurcation of the platelet population into the two phenotypes further creates debate regarding intrinsic versus extrinsic functional programming[8]. Allied to this, subjects with reduced GPVI levels showed reduced thrombus formation[28], implicating platelet heterogeneity with increased activity by platelets with elevated GPVI levels[29]. Overall, a complex picture is emerging, with precision methods required to accurately delineate subpopulations and their functional roles to inform our understanding of platelet interactions governing thrombus formation.

The paracrine signalling inherent to platelet activation represents a technical challenge for measuring single platelet behaviour without interference by the secretion products of activated platelets in the vicinity. This implies the requirement for confinement, discretising the analysis into single platelet measurements. The other requirement is throughput to effectively resolve the functional structure of the platelet population. Droplet microfluidics allows the reliable production of monodisperse droplets in the nanolitre to femtolitre range and has emerged as a powerful tool for packaging single cells in high throughput[30–34]. Here, the surfactants assembled at the aqueous:oil interface prohibit exchange between other aqueous compartments to eliminate platelet–platelet cross-talk. Droplet-based analytical methods have been effectively applied to cell phenotyping[35–39] and are also popularly used for single-cell sequencing[40–43]. In this contribution we describe the first application of droplet microfluidics for mapping the functional behaviour of platelet populations with single platelet resolution. Comparing the responses with bulk platelet populations demonstrates the existence of hypersensitive platelets which can coordinate system-level sensitivity gains, a feature shown to drive heterotypic system polarisation during dual agonist stimulation.

## Results

Microfluidics is suited for the handling of blood cells that naturally exist in a suspension state. This is especially relevant for platelets, which are sufficiently small to be near-neutrally buoyant allowing sustained delivery to the microfluidic device without the need for stirring and associated shear effects[44]. Indeed, platelets are characteristically shear-sensitive[45,46] and the droplet generation junction introduces shear conditions, albeit short-lived (~50 μs). Critically, platelet activation was minimal or absent in the vehicle control samples (Supplementary Fig. 1) demonstrating that the shear conditions for droplet generation, and droplet transport[47], as well as the surfactant and fluorinated PDMS channel walls do not activate platelets.

The experimental concept is illustrated in Fig. 1a along with consideration of the Poisson distribution in Fig. 1b which informs the choice of droplet volume and/or platelet concentration required for the efficient encapsulation of single platelets. For a platelet concentration of $25 \times 10^6$/mL and with further on-chip dilution (×5) with agonist and antibody volumes, this indicates that an 8 pL droplet volume (ø25 μm) produces effective single platelet encapsulation: 3.38% of droplets contain a single platelet and 0.08% contain multiples with a single to multiple ratio of 42. The droplet microfluidic circuit used in this study is shown in Fig. 1c and was used to generate 25-μm-diameter droplets (Fig. 1d, e) at 10.4 kHz for single platelet packaging (352 Hz). This allows >100,000 platelets to be encapsulated in the 5 min collection timeframe. To demonstrate the Poisson distribution effect, high platelet input concentrations were used to observe the relationship between singlet and multiple occupancy events (Fig. 1f). Coupled with the kHz measurement capabilities of flow cytometry the analytical pipeline enables the functional variety of large-scale platelet populations to be readily mapped. The complete sampling to microfluidics and flow cytometry methodology is illustrated in Supplementary Fig. 2.

To evaluate single platelet sensitivity differences, a dose response experiment involving stimulating droplet-confined single platelets with convulxin (a GPVI receptor agonist) was undertaken and compared with the stimulation of platelet collectives. Using $\alpha_{IIb}\beta_3$ activation (inside-out signalling) as the analytical end-point the platelet collectives produced a sigmoidal response curve emerging at 0.1 ng/mL and saturating at 1 ng/mL concentrations. The signal intensity distribution of the collective population indicates continuous functional variety. In comparison, a higher activation threshold is evident with singularly stimulated platelets, with activation emerging at 1 ng/mL and saturating at 10 ng/mL levels (Fig. 2a). The similar signal distribution to platelet collectives indicates that droplet confinement does not downregulate platelet activation. Extending the analysis to a different pathway, the P-selectin exposure end-point for alpha granule secretion, the same increased activation threshold for singularly stimulated platelets was observed (Fig. 2b). Activation and aggregation density plots for platelets stimulated at 3 ng/mL are shown in Fig. 2c and highlight the hypersensitive behaviour of the collective response, the correlation between the two endpoints and the bimodal distribution for singularly stimulated platelets undergoing population-level transition. Importantly, the hypersensitive subpopulation was not observed by platelet collective dilution (up to a further 100-fold dilution), demonstrating the merit of the droplet microfluidics approach for single platelet analysis. To measure the significance in the response differences between single and collective platelets, the relative risk was considered (Fig. 2d, e). At low and high agonist concentrations the relative risk score is ~1.0, showing no effect, and at 3 ng/mL rises to 53 for $\alpha_{IIb}\beta_3$ activation and 6 for P-selectin exposure endpoints, highlighting the significantly ($p$ value $<1 \times 10^{-5}$) distinct hypersensitivity of collectively stimulated platelets.

During confined platelet stimulation with convulxin degranulation results in the accumulation of stimulatory molecules in the droplets and this may lead to enhanced activation (Supplementary Fig. 3). This observation deserves confirmation using

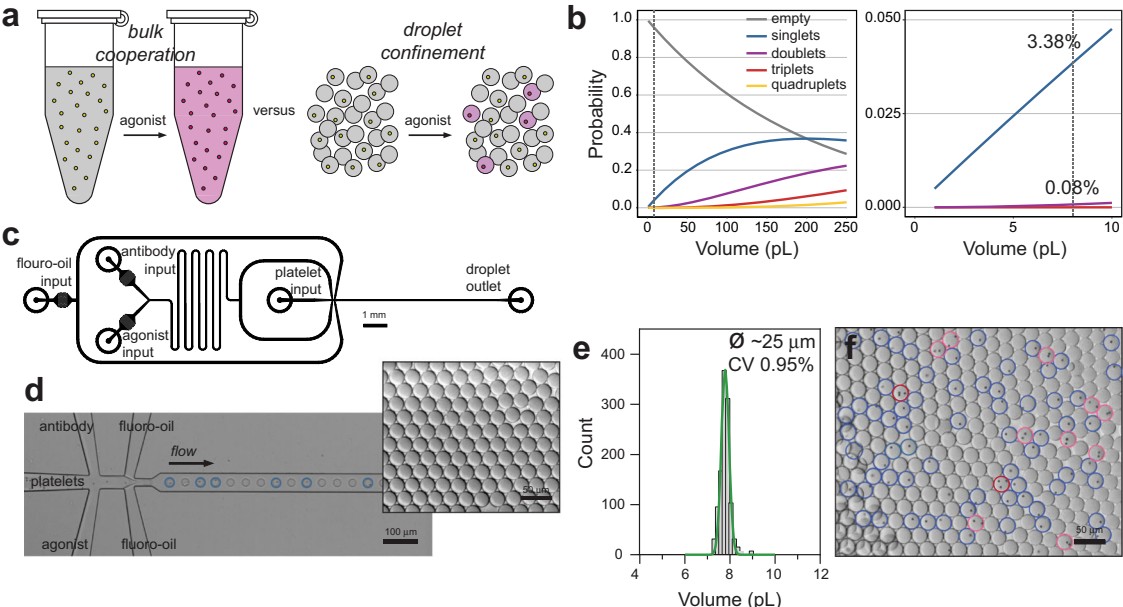

**Fig. 1 Concept and methodology.** Platelet populations cooperate during bulk perturbation experiments to produce an ensembled response whereas droplet compartmentalisation prohibits paracrine signalling to enable single platelet sensitivity measurements (**a**). The Poisson distribution is used to determine an optimal droplet volume for single platelet packaging with minimal multiple platelets (**b**). The microfluidic circuit for combining agonists and antibodies with platelets immediately before droplet generation is drawn to scale (3:1) (**c**). High-throughput (10 kHz) droplet generation and single 2 μm particle packaging (blue circles) (**d**). Inset, droplet monodispersity is indicated by hexagonal packaging. Microfluidic conditions produce ~8 pL (CV ± 0.95%) droplets (**e**). Poisson distribution impacting encapsulation illustrated using an excessive, 125 × 10⁶/mL, 2 μm platelet-sized particle input concentration (**f**); singlet occupancy (blue; 23.9%), doublets (pink; 3.6%) and triplets (red; 0.7%).

inhibitors but nevertheless at the activation transition with a 3 ng/mL convulxin stimulation in droplets a clear bimodal distribution is evident with the activated population having a higher $\alpha_{IIb}\beta_3$ activation signal than platelet collectives also undergoing activation transition (0.3 ng/mL).

The sensitivity gains emerging from collective platelet behaviour were reproducible, with equivalent dose responses, both single and collective, obtained from the same donor three times over a 9-month period (Supplementary Fig. 4). When the study was extended to a cohort of eight healthy yet diverse donors (gender, age, BMI, smoking) the same pattern was observed, confirming the generality of the hypersensitive collective response, and allowing an efficacy model to be generated. For both $\alpha_{IIb}\beta_3$ activation and P-selectin endpoints, the collective convulxin response had an $EC_{50}$ value of 0.4 ng/mL, whereas the single platelet $EC_{50}$ was 7.5 ng/mL (Fig. 2f, g). The 19-fold median sensitivity gains demonstrates the importance of hypersensitive platelets and their cooperative influence.

To confirm that the molecular $\alpha_{IIb}\beta_3$ activation and P-selectin endpoints represent functional behaviour the dose response study was extended to larger droplets (65 pL; ø50 μm) packaging 0–15 platelets. Here the presence of hypersensitive platelets was predicted to result in aggregation at moderate convulxin concentrations. At low concentrations (0.01 ng/mL) P-selectin negative platelets are observed as multiple, spatially distinct entities within each droplet. At moderate concentrations (1 ng/mL) this droplet case is observed along with the other case in which droplets contain a single platelet aggregate. These stain positive for P-selectin are typically large and, dictated by the Poisson statistic, must generally contain multiple, co-localised platelets (Fig. 2h). Overall, this points to the existence of hypersensitive platelets in a large fraction of droplets when treated with 1 ng/mL convulxin. At maximal concentrations (100 ng/mL) all droplets contain platelet aggregates. Plotting the flow cytometry

data shows a closer similarity with the collective platelet response (Fig. 2i). However, a distinct bimodal distribution still results when using 1.0 ng/mL convulxin. Elevated P-selectin signals in droplets relative to collective conditions are observed at 10 and 100 ng/mL convulxin. This could indicate autocrine and paracrine signalling resulting from the accumulation of degranulation products within the droplets, but again requires confirmation using inhibitors. This experiment demonstrates the functional consequence of broad-spectrum sensitivity with cooperation and that minimalistic platelet cooperation models can be used to understand transition states and the linkage between probabilistic molecular events and collective functional outcomes.

Collective sensitivity gains are attributed to the existence of low abundance hypersensitive platelets which, upon activation, degranulate to activate platelets in the vicinity that were insensitive to the initial stimulus. These modes of paracrine signalling produce a spatiotemporal corralling effect that drives platelet cooperation to deliver the collective response. Nevertheless, sufficient numbers of activated platelets are required to polarise the entire platelet population into an activated response (e.g. Fig. 2a; collectives with 0.1 ng/mL convulxin). Our experiment involved platelets diluted to approximately 1/50th of in vivo concentrations, suggesting digital activation may well occur under physiological conditions with insufficient volume to disperse paracrine signals. Platelet cooperation is mediated through the secretion of alpha granules as evidenced by P-selectin exposure, but also ADP and serotonin secretion from dense granules as evidenced by CD63 presentation (Supplementary Fig. 5), a marker for dense granule and lysosome fusion with the membrane[48]. The dense granule secretion pathway had a higher activation concentration than the alpha granule secretion pathway, although upon activation the kinetics of dense granule secretion are faster[49] which may allow the timely augmentation of pathways for specialised platelet activation[50–52].

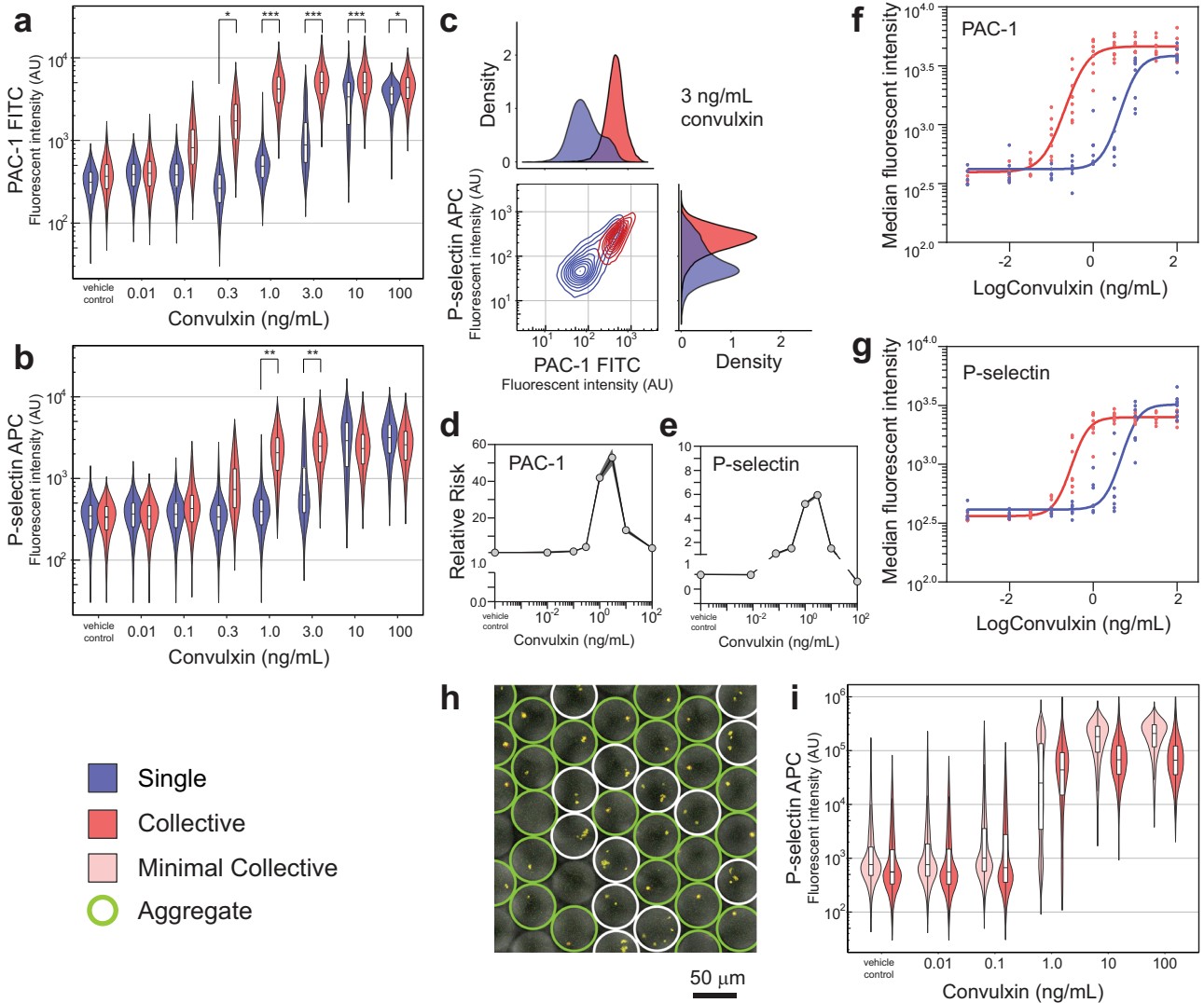

**Fig. 2 Broad-spectrum response to convulxin stimulation and hypersensitive collective behaviour.** Violin plots comparing the activation of single platelets with platelet collectives using a convulxin dose response experiment, with PAC-1 binding to activated $\alpha_{IIb}\beta_3$ (**a**) and P-selectin exposure (**b**) endpoints (relative risk; *>2, **>5, ***>10). Contour plot and density plots of the emergence of hypersensitive single platelets at 3 ng/mL convulxin concentrations, while the collective population is fully activated (**c**). Relative risk analysis was used to determine the significance of the ~20-fold differences between the single and collective platelet responses using PAC-1 (**d**) and P-selectin (**e**) endpoints with confidence intervals determined by the Koopman asymptotic score. The $E_{max}$ model was used to show a consistent difference between single and collective platelet behaviour across a diverse cohort (age; gender; smoking; BMI; exercise) of eight healthy donors using PAC-1 (**f**) and P-selectin (**g**) endpoints. Droplet volume scaling to 65 pL produces minimal collectives (0–15 platelets with ~500 × 10^6 platelet/mL inputs) to allow aggregation responses to be investigated. Triple fluorescent imaging (P-selectin, CD63 and CD42b) with brightfield overlay of minimal platelet collectives stimulated with 1 ng/mL convulxin. Droplets containing aggregates are indicated by a green ring and those with multiple separate platelets by a white ring (**h**). Resulting dose response violin plots of minimal platelet collectives compared with bulk platelet collective responses (**i**). To measure aggregates by flow cytometry, doublet(+) gating was removed, thereby increasing the signal spread by the inclusion of various aggregate scales (2–15 in the case of minimal collectives) along with signals from active single platelets and inactivate single platelets. For each single platelet condition, $n = 10,000$–36,000 platelet events were measured, and $n \approx 48,000$ for the collective conditions.

The study was extended to other agonists; the peptide TRAP-14 functional motif was used in place of thrombin to activate the PAR-1 receptor and as before $\alpha_{IIb}\beta_3$ activation and P-selectin exposure endpoints were measured. The median activation threshold was again increased for single platelets stimulated in droplets, indicating that coordination by low abundance hypersensitive platelets reduces the activation threshold for platelet collectives. The emergence of a bimodal population distribution with singularly stimulated platelets was also observed for both endpoints at 12.5 and 25 μM TRAP-14 concentrations (Figs. 3a, b). The sigmoidal dose response again signifies continuous sensitivity variation. A small yet sensitive (~4-fold) subpopulation of

single platelets stimulated with 12.5 μM TRAP-14 is evident. Experiments with the weak agonist ADP[50], stimulating the P2Y₁ and P2Y₁₂ receptors showed minor sensitivity gains, with a small but hypersensitive single platelet subpopulation identified from droplets at 0.1 μM ADP concentrations (Supplementary Fig. 6).

The collective sensitivity gains are agonist dependent, scaling with convulxin>TRAP-14>ADP and correlating with the potency of the agonist[19]. This implies that collective sensitivity gains are most advantageous for triggering thrombus formation upon stimulation with collagen. Overall, collective sensitivity gains represents a strategy for robust, consensus-level, homoeostasis emerging from paracrine cooperativity. This also conveniently

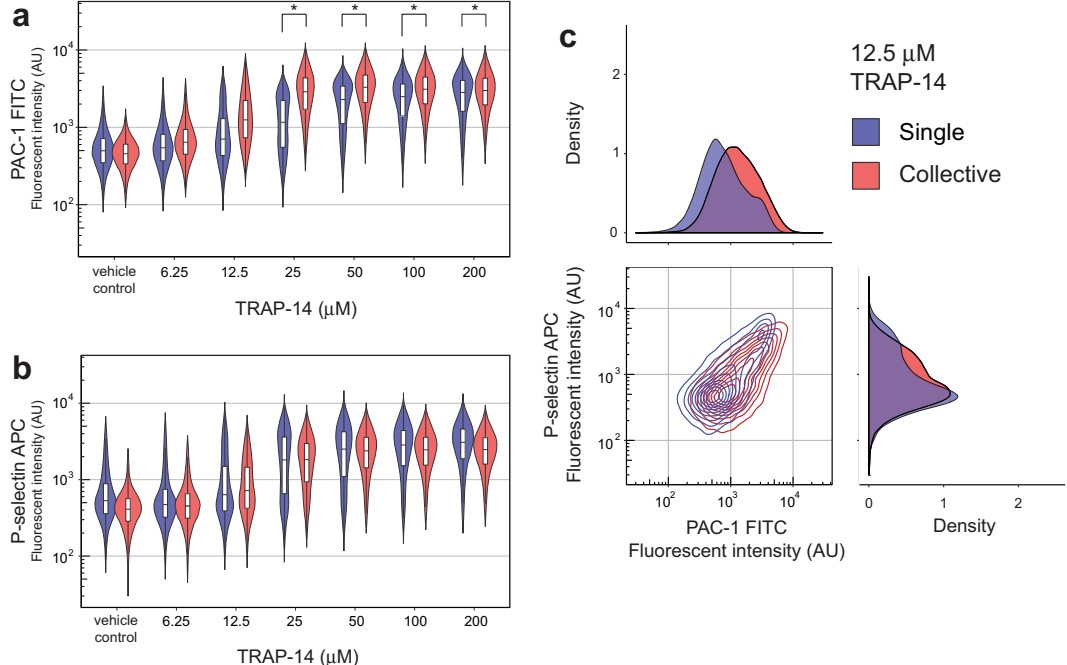

**Fig. 3 Variable TRAP-14 response and increased collective sensitivity.** Violin plots comparing the activation of single platelets with platelet collectives using a TRAP-14 dose response experiment, with PAC-1 binding to activated $\alpha_{IIb}\beta_3$ (**a**) and P-selectin exposure (**b**) endpoints (relative risk; *>2). Contour and density plots showing increased collective sensitivity and the emergence of the more sensitive single platelet subpopulation with a 12.5 μM TRAP-14 stimulation (**c**). For each single platelet condition, $n = 7000$–22,000 platelet events were measured, and $n \approx 48,000$ for the collective conditions.

exploits inherent platelet variability, thereby bypassing the need for functionally uniform platelets. Whether this diversity model involving community cross-talk for the transition from a dispersed state to localised recruitment and responsiveness can be generalised to other scenarios such as immune infiltration remains to be seen.

Functional variety is a common feature of cellular systems enabling powerful system responsiveness and control. This research shows that broad and continuous sensitivity distributions of single platelets interfaced via paracrine signalling produces robust collective sensitivity gains. During dual stimulation with collagen and thrombin, platelets are known to polarise into two distinct populations: pro-coagulant and pro-aggregatory phenotypes[22]. The intrinsic or extrinsic nature of this heterogeneity is a matter of debate[8]. Again using droplet confinement we sought to resolve this debate and also to question the role of collective hypersensitivity in the emergence of the heterotypic response.

A dual stimulation dose response experiment was undertaken requiring the addition of calcium to unwashed platelets to enable membrane inversion for annexin V binding to phosphatidylserine moieties and also the addition of rivaroxaban to prevent additional thrombin formation by factor Xa and gly-pro-arg-pro (GPRP) to limit the formation of fibrin fibres. The responses of single and collective platelet populations are compared using violin plots (Supplementary Fig. 7) and show that that the different conditions and integration of the two activation pathways alters system sensitivity (see Fig. 2a), with platelet collectives pro-coagulant (annexin V high; $\alpha_{IIb}\beta_3$ low) and pro-aggregatory (annexin V low; $\alpha_{IIb}\beta_3$ high) heterotypic states emerged with a 100 ng/mL convulxin and 1.0 U/mL thrombin stimulation and is consistent with the literature[16,17]. At the same concentrations droplet-confined, single platelet populations do not fully polarise, with an unresponsive third population (Fig. 4a). Again this demonstrates the need for cooperation to enhance system sensitivity to activate all platelets. At higher dual stimulation doses (300 ng/mL convulxin and 3.0 U/mL thrombin) single platelet

populations fully polarise into pro-coagulant and pro-aggregatory states. By excluding paracrine cross-talk, this confirms the intrinsic origins of heterogeneity. Indeed, removal of paracrine cooperative effects produces a fully digital pro-coagulant or pro-aggregatory response (Fig. 4b). Importantly, these findings are made possible by single platelet confinement, advocating the use of droplet microfluidics to accurately delineate intrinsic single platelet phenotypes. In contrast to droplet-confined stimulation, the heterotypic distribution of platelet collectives involves some platelets with graded intermediate states. This implies the role of extrinsic effects for the generation of subtler phenotypes likely required to enable more sophisticated functionality throughout the thrombus.

In this study, collective sensitivity gains are shown to be a general feature of human platelet biology. To gain further insights into this behaviour, these platelets were gated for characterisation (Supplementary Fig. 8). Their forward and side scatter properties are indistinguishable from other, insensitive platelets. The CD42b signal (monomer component of the Von Willebrand factor receptor, GPIb-IX-V) for the hypersensitive platelets is similar albeit slightly reduced as a consequence of matrix metalloproteinase excision upon activation[53]. Further investigations involving large-scale antibody panels for highly multiplexed cytometry or more global proteomic[54] and even transcriptomic screens[55,56] following platelet sorting will be needed to determine the composition of the hypersensitive platelet subpopulation. While this holds promise for the identification of elements governing system behaviour and potential hubs for therapeutic intervention, caution is required for extension to in vivo contexts in which multiple cues integrated in time and space mediate responses that allow highly robust homoeostatic control.

## Discussion
In summary, the microfluidics and cytometry methodology was used to identify a broad-scale sensitivity continuum containing hypersensitive platelets which coordinate collective sensitivity gains

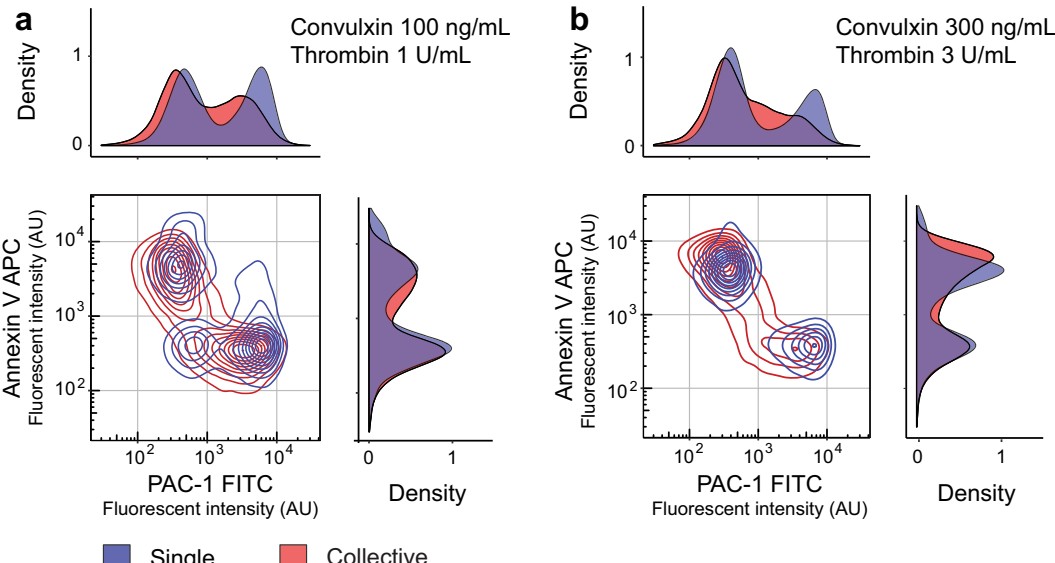

**Fig. 4 Intrinsic heterotypic states in response to dual stimulation.** Stimulation of platelet collectives with 100 ng/mL convulxin and 1 U/mL thrombin produces pro-coagulant (annexin V high; $\alpha_{IIb}\beta_3$ low) and pro-aggregatory (annexin V low; $\alpha_{IIb}\beta_3$ high) states. With the same stimulation, single platelets produce a third unresponsive population (annexin V low; $\alpha_{IIb}\beta_3$ low), indicating the requirement for paracrine cooperation to achieve complete population activation (**a**). Single platelets stimulated with higher 300 ng/mL convulxin and 3 U/mL thrombin concentrations drives platelets exclusively to functionally distinct pro-coagulant or pro-aggregatory states (**b**). Cooperation in platelet collectives at both dual stimulations concentrations directs some platelets into graded, intermediate activation states. For each single platelet condition, $n = 14,000–26,000$ platelet events were measured, and $n \approx 48,000$ for the collective conditions.

by paracrine cooperativity to produce a robust system response. This feature can drive system polarisation into pro-aggregatory and pro-coagulant states during dual stimulation. The understanding that collective platelet dynamics are dependent upon a hypersensitive subpopulation could be of critical importance when considering that imbalance in the platelet population structure represents a potential route to pathology, either bleeding or arterial thrombosis leading to heart attacks and strokes.

Different perspectives require evaluation to fully interpret our findings: In principle, hypersensitivity could arise from natural variation within a functionally homogeneous platelet population or may be an important characteristic of a functionally distinct subpopulation. In the former case, all platelets are essentially equivalent in their functional responses, yet if platelet activation is an inherently stochastic process then the time-to-activation of each individual platelet will be described by a random variable. In this case, the observed variation in response may arise as a consequence of this underlying temporal stochasticity rather than being due to functional variability in the population per se. Similar stochastic mechanisms have been shown to be important in generating functional heterogeneity in other contexts[57], for example within stem cell populations[58]. Alternatively, hypersensitive platelets may comprise a genuinely functionally distinct subpopulation. In the classic case of the emergence of bacterial resistance to virus infection, Luria and Delbrück[59] applied a fractionation technique with modelling to determine that stochastically acquired mutations, not a pre-existing subpopulation, produced resistance. We anticipate that such a combination of experimental precision with single-cell resolution and mathematical models[60] will help resolve this issue. In doing so, we will be a step closer to identifying system-level prognostic biomarkers and designing new therapeutic interventions.

## Methods
**Device design and fabrication**. Microfluidic channels were 20 μm in height and with a width of 22 μm at the droplet generation junction for the reproducible generation of 25 μm (~8 pL) droplets. The complete device design is available in the

Supplementary Information (Supplementary Data 1), and involves separate agonist and antibody inlets that combine with the platelet inlet in advance of interfacing with the fluoro-oil phase at the droplet generation junction. All inlets, excepting the platelet inlet, included filter structures to remove any particulate and fibre contaminants during droplet formation. Microfluidic devices were prepared by standard SU-8 photolithography followed by poly(dimethylsiloxane) (PDMS, Sylgard 184) to polyurethane (Smooth-On 310) mould cloning for parallel replication by soft lithography in PDMS at 60 °C for 2 h. Inlet/outlet ports for plug and play interconnection were produced using a 1-mm-diameter Miltex biopsy punch (Williams Medical Supplies Ltd). Devices were bonded to PDMS-coated glass microscope slides using a 30 s oxygen plasma treatment (Femto, Deiner Electronic) followed by surface functionalisation using 1% (v/v) trichloro(1*H*, 1*H*, 2*H*,2*H*-perfluorooctyl)silane (Sigma Aldrich) in HFE-7500™ (3 M™ Novec™). Minimal platelet collectives were encapsulated in 50-μm-diameter droplets, generated with a 50-μm high microfluidic device with a 50-μm-wide droplet generation junction that were fabricated as described above.

**Participants and sampling**. Blood was obtained from healthy volunteers after obtaining informed consent under institute (ERGO 5538) and South Central-Hampshire B (REC: 14/SC/0211) ethical approvals. Participants were free from anti-platelet medication, such as aspirin for 2 weeks and >24 h free from other non-steroidal anti-inflammatory medication. Venepuncture with a 21G needle was used to collect blood in vacuum tubes containing 1:10 v/v 3.2% trisodium citrate (first 4 mL discarded). Tubes were gently inverted, centrifuged at 240*g* for 15 min without brake to prepare platelet-rich plasma (PRP) that was rested for 30 min prior to experiments. Platelet counts were determined using the method described by Masters and Harrison[61], involving a CD61 antibody and an Accuri C6 instrument (BD Biosciences). Platelets were subsequently diluted to a concentration of $25 \times 10^6$/mL in HEPES buffer (136 mM NaCl, 2.7 mM KCl, 10 mM HEPES, 2 mM MgCl$_2$, 0.1% (w/v) glucose and 1% (w/v) BSA (pH 7.45)) for dose response experiments.

**Droplet microfluidics**. Medical grade, sterile polythene tubing (ID 0.38 mm; OD 1.09 mm) was used to directly interface syringes with 25G needles to the microfluidic ports. Syringe pumps (Fusion 200, Chemyx) were used to deliver reagents. The Poisson distribution effect was evaluated using NIST, monodisperse 2-μm-diameter polystyrene particles (4202A, ThermoScientific™). Platelet experiments involved the delivery of HFE-7500 fluoro-oil (3M™ Novec™) with 0.75% (v/v) 008-fluorosurfactant (RAN Biotechnologies) at 20 μL/min, antibody and agonist solutions at 2 μL/min and platelets at 1 μL/min to generate 25-μm-diameter droplets. High-speed imaging (2500 fps) using a Miro eX2 camera (Phantom) mounted on an open instrumentation microscope (dropletkitchen.github.io) was used to document droplet generation and an inverted fluorescent microscope (CKX41, Olympus) fitted with a QIClick camera (Teledyne, QImaging) was used to image

droplet contents. Droplets were collected for 5 min, incubated while resting at room temperature in the dark for 10 min, then combined with CellFix fixative (BD Biosciences) and subsequently with 1*H*, 1*H*,2*H*, 2*H*-perfluoro-1-octanol (PFO, Sigma Aldrich) to destabilise the droplet interface and break the emulsion with minimal platelet losses during extraction of the aqueous volume containing fixed platelets. The 5 min continuous droplet collection period followed by a 10 min incubation produced overall droplet incubation times ranging from 10 to 15 min that were optimal for distinguishing activated platelets from the vehicle control platelets. Incubations can be extended to 60 min while retaining appreciable signal to noise cytometry data.

The collective experiments were undertaken in microcentrifuge tubes and involved matched conditions to the droplet experiments, in which $25 \times 10^6$/mL platelet samples were diluted fivefold by the addition of equal volumes of agonist and antibody reagents and incubated for 15 min before fixation. In the case of the 50 μm droplets, the reagent flow rates were 80 μL/min for fluoro-oil, 4 μL/min undiluted platelet rich plasma ($\sim5 \times 10^8$/mL), and 8 μL/min for convulxin and antibody inputs. In the absence of stirring, platelets were incubated for 60 min prior to emulsion breaking and fixation to allow aggregation to be concluded.

**Flow cytometry**. Platelets were stimulated with convulxin (Enzo Life Sciences), a snake venom which activates the GPVI receptor, TRAP-14 (Bachem AG) an agonist of the PAR-1 receptor or ADP (Sigma Aldrich) an agonist of the P2Y$_{12}$ receptor. The dual agonist experiment involved stimulation with convulxin and thrombin (Sigma) in the presence of 2.5 mM CaCl$_2$. Here, coagulation was prevented using 0.5 μM rivaroxaban (Advanced ChemBlocks Inc.) and 100 mM H-Gly-Pro-Arg-Pro-OH (GPRP, Bachem) added to the HEPES platelet dilution buffer. Fluorescent antibodies and selective stains were used to detect biomarkers: fluorescein isothiocyanate (FITC)-conjugated PAC-1 (PAC-1 clone at 1.25 ng/μL), allophycocyanin (APC)-conjugated CD62P (P-selectin, AK-4 clone at 0.63 ng/μL), FITC-conjugated anti-CD63 (H5C6 clone at 2.0 ng/μL), R-phycoerythrin (PE)-conjugated CD42b (HIP1 clone at 1.25 ng/μL) and Annexin V at 0.08 ng/μL were obtained from BD Biosciences. Following treatments, antibody incubation and fixation, samples were diluted in PBS and measured using an Accuri C6 flow cytometer (BD Biosciences). Platelets were identified using a gate on CD42b-PE intensity, with doublets and non-platelet-sized events removed by gating. An overview of the gating procedure is provided in Supplementary Fig. 9. Importantly, platelet dilution and short 15 min incubations without stirring of the collective samples did not increase the gating out of platelet doublet/aggregate events, thereby surveying the population without bias. In the case of the 50 μm droplets, CD42b-PE, CD63-FITC and CD62P-APC antibodies were used and incubated for 60 min prior to fixation and emulsion breaking. Here, doublet gating was not applied to retain platelet aggregates in the analysis and PAC-1 staining was not used to avoid interference with aggregation.

**Statistics and reproducibility**. Droplet images were analysed using ImageJ (NIH) and flow cytometry data using R. All flow cytometry data were tested for normality using the Shapiro–Wilk test (GraphPad Prism). To compare single and collective platelet responses from a single donor, the relative risk statistic was used to quantify the association between stimulation and response (R, epitools). Violin plots were made using the 1–99 percentile (R, ggplot2). The overall cohort response difference between single and collective platelets was plotted using an efficacy maxima ($E_{max}$) sigmoidal model generated in GraphPad Prism. The cohort was diverse, with five male and three female volunteers, aged between 20 and 60 and with one smoker. To ascertain reproducibility, some donors were measured three or more times. For the single platelet flow cytometry analyses, an average of $\sim$30,000 platelets events were measured for each condition and $\sim$48,000 platelet events for each of the collective platelet conditions.

**Reporting summary**. Further information on research design is available in the Nature Research Reporting Summary linked to this article.

## Data availability

The raw cytometry.csv files are made available via FigShare (https://doi.org/10.6084/m9.figshare.12086103.v1).

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

## Acknowledgements

We are indebted to the blood donors and we thank Simon Lane for support with ImageJ analysis, Johan W.M. Heemskerk for critical manuscript feedback and Joanna D. Stewart for proofing the manuscript. The research was funded by the Marie Curie (333721, to J.W.), the British Heart Foundation (FS/13/67/30473, to M.S.A.J.) and the Medical Research Council (MC_PC_15078, to M.S.A.J.).

## Author contributions

J.W. conceived the project; M.S.A.J. did the experiments and analysed the data; N.A.E., B.D.M. and J.W. supervised the research; J.W. wrote the manuscript and M.S.A.J., N.A.E. and B.D.M. reviewed the manuscript and approved the final version.

## Competing interests
The authors declare no competing interests.
