## [Peer Review File · Communications Biology]

Reviewers' comments:

Reviewer #1 (Remarks to the Author):

In this report, the authors describe a novel method to isolate single platelets: droplet microfluidics. This method enables the characterization of single platelets and the investigation of individual platelet responses upon activation. The authors show that hyperreactive platelets within the total platelet population enhance the collective platelet response by paracrine signaling, thereby lowering the agonist concentration needed for maximal activation.

The authors present a methodology that is relevant for the further characterization of the individual platelet response. Droplet microfluidics can be a useful and relevant tool in future research towards platelet heterogeneity and platelet populations. I do however have some questions for the authors.

1. When looking at the vehicle controls of the single platelets, there appears to be a substantial number of platelets that have a relatively high fluorescence signal for PAC-1 and P-selectin labeling (Figure 3A-B, Suppl Fig 2 and Suppl Fig 5). This fraction is smaller in the collective response. The authors state that platelet activation was absent in the vehicle control, but can the authors exclude pre-activation/low-grade activation and desensitization of the platelets due to droplet generation. It would be helpful when the authors can include data on platelet activation of platelets that have been passed through the droplet microfluidics device without actual droplet generation to show that droplet generation does not affect platelet reactivity. Further, the description of the preparation of the collective samples (platelet count, dilution) and the protocol used to assess platelet activation in the collective samples, are unclear.

2. The authors used the generation of larger droplets containing multiple platelets to show that PAC-1 and anti-P-selectin labeling are a good representation of platelet activation. They observe that platelets within the larger droplets -called minimal collectives- aggregate at moderate and maximal concentrations of convulxin (Figure 2H). However, when aggregation occurs, flow cytometry will not provide an accurate estimation of the platelet activation state (Figure 2I). Can the authors please comment?

Further, incubation with the antibodies was for 60 minutes. Why did the authors choose for 60 minutes incubation in the case of 50 μm droplets/minimal collectives in comparison to 10-15 minutes incubation for the smaller droplets/single platelets?

3. The generation of single platelets using the droplet microfluidics device offers the opportunity to investigate the contribution of autocrine signaling to the individual platelet response. The authors also state that autocrine molecules inside the droplets enhance platelet activation. Have the authors looked at the effect of inhibition of the autocrine/paracrine TXA₂ and ADP pathways in single platelets vs. platelet collective samples upon convulxin and TRAP stimulation?

4. The contribution of paracrine signaling to the overall platelet response becomes also apparent from the experiments where ADP is used as an agonist. The authors have chosen concentrations of ADP ranging from 0.01 to 10000 μM . Why were such high concentrations of ADP (1000-10000 μM) used? Can the authors explain the decrease in integrin activation and P-selectin expression?

5. The authors show that dual stimulation of single platelets leads to the formation of an unresponsive third population of platelets (Figure 4A). However, stimulation of single platelets with the same concentration of convulxin (100 ng/mL) does result in full integrin activation after 15 minutes (Figure 2A). It would be interesting to see whether the third unresponsive population is only unresponsive towards annexin binding and was positive for PAC-1 binding at earlier timepoints. A time series

experiment would clarify this.

Minor comments:

Figure 2A-C: the range of the fluorescence intensity is different compared to the other figures in the manuscript and supplement (range: 10^{1-3} compared to 10^{2-4} in other figures).

Reviewer #2 (Remarks to the Author):

Jongen et al have applied microfluidic isolation of single platelets. As far as I know, this is a first time and an important step in trying to understand if, why and how the normal platelet population is composed of (functional) subpopulations.

Platelets are small but very abundant cells, roughly 5% of all cells in the human body are platelets (Sender et al, Cell 2016;164(3):337-40). Platelets are highly complex cells, on the morphological and functional level. During blood circulation, platelets serve homeostasis by passively and actively surveilling the vasculature. Integrity breaches are effectively sealed off by hemostasis, including an indispensable role for platelets. Platelets may further contribute to tissue healing as well, although that function is less understood.

Jongen et al have now used single cell isolation to investigate (possible) platelet heterogeneity, in vitro. In particular the authors have assessed the response of single platelets to increasing concentrations of known agonists. As an outcome measure the authors have used known reporter molecules expressed on the (activated) platelet surface using flow cytometry. The research question(s), the study design, the methodology and the written manuscript is strong. The data are very interesting and will advance the field and/or trigger new questions. The data furthermore lead to original conclusions. The data in Figure 2A-G in particular suggest that bulk platelet activation is "led" or "instigated" by a rare subpopulation of platelets that provides sensitivity gains to the bulk. These data stand out as the key experiment of the manuscript. I have a couple of remarks.

1. From the data in Figure 2A, I conclude that the 'hypersensitive' platelet subpopulation must be extremely rare. The reason for this conclusion is that the first resolution of this population in single cell analysis is technically only possible at 3 ng/mL of CVX. At this CVX concentration, bulk platelet preparations are fully activated. If these 'hypersensitive' platelets indeed sensitize the platelet bulk then they must have been present at 0.3 ng/mL as well (in bulk) while at that concentration not yet distinguishable in single cell analysis. At 0.3 ng/mL CVX the 'hypersensitive' platelets are thus either in the noise or just not present in quantities that allow resolution in flow cytometry, i.e. rarer than 1/100,000 assuming all events were read. If time proves the investigators right, then this concept is a landmark novelty. But the authors wisely do not claim that the 'hypersensitive' subpopulation is required for successful bulk activation even though their first sentence at L223 comes close. Therefore, this reviewer remains critical to the possible translation of these findings in vivo. One reason is that thrombocytopenia to low levels of normal (up to 5-8% or 20,000 platelets per μL) generally does not cause bleeding in otherwise healthy humans, meaning that the very rare subpopulations identified in this manuscript and seemingly important to sensitize bulk platelet activation would become even rarer (or almost non-existing) in these patients. Proving relevance of these findings to hemostasis/thrombosis is therefore challenging. The authors may want to briefly mention possible translational hypotheses in order to provide context and trigger the community in designing experiments that will address this.

2. The authors did well in designing a system that manages high throughput (10kHz) droplet

formation. Yet, the first collected single platelet still experiences a 5 minute longer dwell time (at least) than the final collected platelet. This 5 minute spread between the first and last platelet is unavoidable but nonetheless constitutes a study limitation. What is the chance that heterogeneity is introduced caused by this time gap? Do the authors agree that this is an experimental limitation? If not, please disprove and if yes, suggestion to mention this limitation in the manuscript.

3. At what platelet concentration were the 'collective' experiments performed? Was that also 25,000 per μL ? Please indicate the platelet concentration in the manuscript.

4. Please mention the total number of (gated) events analyzed by flow cytometry in both conditions (single vs collective)?

5. For the 'collective' experiments using bulk activated platelets, were measures taken to avoid platelet aggregation (Tirofiban? Reopro?)? Were platelet 'aggregates' found in these samples? What could possible artifacts be for the flow cytometry readout if (partial) aggregation was taking place?

6. Please add "per mL" in Supplementary figure 1 below the syringes, in the first bullet, so to indicate platelet concentration instead of the unitless measure now.

7. L31 "a state of decay" is not what references 14,15 demonstrate. Maybe the authors intend to say that from the typical steady state of platelet bioproduction (IN) and clearance (OUT) automatically follows a heterogeneity in terms of senescence for any bulk platelet preparation isolated in a theoretical snapshot? Suggest to rephrase in order not use "decay".

8. L35 it's probably not necessary to refer to this many papers for COAT platelets (ref 22-24). Suggestion to retain one, preferably the milestone primary report.

9. L64 "platelet activation was absent in the vehicle control samples". This seems correct from the data, but I had to look for it when reading this sentence. Suggestion to refer to a Figure here. A figure that contains these 'validation' data. Other reviewers/readers may not require this, but I found it confusing because the statement is important but seemingly without data when not referred to a figure. Only later on in the manuscript becomes clear that 'untreated' controls are included in every experiment.

10. L130-L141 is not a very strong paragraph in the paper, my remarks below may assist in making it stronger or have the authors decide not to keep it. In my opinion this paragraph does not add much to the overall finding.

11. L133-135 This sentence is not supported by data "single aggregates of platelets are observed in droplets containing hypersensitive platelets, and not in droplets without hypersensitive platelets". There is no panel in Figure 2 that supports this statement.

12. L134-L135 "At maximal concentrations (100ng/mL) all droplets contain platelet aggregates (Figure 2H)" needs more explanation or is poorly supported by the data. The labeling of droplets in panel H is green, white or unlabeled. Does this mean something? The outcome measure "aggregation" is not well described and therefore hard to interpret. In addition, the figure caption mentions 1ng/mL and not 100ng/mL. Please clarify, suggestion to amend.

13. L138-139 "This is indicative of autocrine signaling" is speculative at this instance. Suggestion to rephrase or to omit from this paragraph. The hypothesis of autocrine signaling is correctly mentioned in the final paragraph of the Results & Discussion section.

14. L145-147 "Nevertheless, sufficient number of activated platelets are required to polarize the entire platelet population into an activated response (e.g. Figure 1A)." Do the authors mean Figure 2A? Figure 1A is a model of the study design. Please check.

15. L147 "our experiment involved platelets diluted to approximately 1/100 of in vivo concentrations." Referral to my earlier remark (3). The platelet concentration reported in the methods section is 25X106 per mL. This is 25,000 per μL or 1/10 of normal, not 1/100. The concentration of one platelet per 8 pL is in fact three times higher than this, on average an equivalent of 125,000 per μL or 1/2 of normal. This needs to be addressed by the authors to avoid confusion and to provide relevance and context because the phrase that follows "suggesting digital activation may well occur" may no longer hold true.

16. L150 The authors should provide a strong reference that demonstrates that CD63 is a reliable

reporter for dense granule secretion. Preferably not a review paper, but an actual original research paper that proves this statement. The CD63 receptor is expressed on platelet granule membranes and its density increases upon sustained platelet activation, but most papers mention this receptor as part of dense and lysosomal granules.

17. L151 "dense granule secretion pathway has a higher activation threshold" seems not to corroborate with reports on dense and alpha granule secretion kinetics in platelets. For instance, Jonnalagadda, D et al Blood 2012;120(26):5209-16 shows that serotonin (dense granule component) is released before PF4 (alpha granule component).

18. L153 "Again, autocrine signaling..." same remark as above (13). It is speculative at this instance. Suggestion to rephrase or to omit. The hypothesis of autocrine signaling is correctly mentioned in the final paragraph of the Results & Discussion section.

19. L165 syntaxis "are", should be "is"

20. L174-L176 "this scaling correlates ...". The authors make a link with a possible in vivo time sequence for hemostasis here. But please provide an original research paper that proves this scaling in vivo. The reason for my skepticism is that during in vivo hemostasis, these separate steps are probably not that strictly time resolved as we tend to think from our in vitro models. Thrombin may be formed early by the tissue factor pathway in the gushing wound prior to significant platelet accumulation onto collagen. Amplification signals (ADP and/or Thromboxane A2) may be secreted from platelets following initial GPIb-VWF interactions (Cannobio I et al Cell Signal 2004;16:1329-44 and Liu J et al Blood 2005;106:2750-56) prior to firm arrest onto collagen. The time resolution described here appears very "textbook", but may be too speculative in this context.

21. L191-L205, Figure 4 and L305-L309 mention the use of GPRP as an inhibitor for coagulation. I understand that some way of inhibiting coagulation is required in this experiment using Ca²⁺. However, GPRP may also influence PS exposure on platelets and influence the data shown in Figure 4 and Figure S6. Check Brzoska et al, PLoS ONE 2013;8(2):e55466 (DOI 10.1371/journal.pone.0055466). Did the authors try using only DOACs (like rivaroxaban or edoxaban or alike)? I don't consider this a major issue, but GPRP is not very specific and may stick or bind causing artifacts to normal platelet activation.

22. L223-236 is a strong and well balanced paragraph, this is a good way to interpret the data.

Reviewer #3 (Remarks to the Author):

The manuscript by Jongen and colleagues describes the use a droplet microfluidic platform to isolate single platelets to investigate the differences between the single cell and collective response of platelets under different forms of stimuli. The authors devised a clever approach using droplet microfluidics to isolate the platelets to prevent paracrine signaling that occurs in a bulk suspension (herein referred to as a collective). This approach allows for the interrogation of single cells in the absence of paracrine signaling. After isolate, the platelets were fixed, stained, and screened using flow cytometry. The authors found that the hypersensitivity of the collective population of platelets was driven by a small hypersensitive subpopulation of platelets by comparing the responses of single and collective cells monitored by integrin and P-selectin activation. The authors postulate that this sensitivity is a function of the concentration and choice of agonist used to stimulate the platelets. Overall, the manuscript is well-written and easy to follow with clear and easy to interpret figures. This reviewer has a few minor comments that the authors should address prior to publication.

1) At the end of the manuscript the authors mention potential temporal control of the system and that the exposure time to an agonist and the length of time a platelet is isolated from the collective could further influence the activation of the selected biomarkers. In the manuscript, the authors only

selected a single time point. Can they provide some insight on why a 10 min incubation time was selected and what they would expect to see if the droplet incubation time was increased or decreased?

2) Related to the above comment, can the authors provide insight on platelet viability in the droplets for the 10 min incubation and potentially greater incubation times? What is the upper threshold of the microfluidic system?

3) Can the authors provide insight on how the droplet breaking procedure affected platelet viability and platelet recover? Do they have any insight on platelet losses and/or changes in platelet response/activation using the system?

4) Can the authors report the total cell numbers analyzed for each of the experiments? It would be helpful to see how many cells were analyzed for the single platelet versus the collective experiments.

5) This reviewer had a difficult time seeing the difference between the single and minimal collective populations in figure 2. Another color for the minimal collective data would make it easier to see the difference.

Response to Reviewers Feedback:
manuscript ID: COMMSBIO-20-0383

15th April 2020

Dear Reviewers,

We found the reviews extremely helpful to clarify and overall improve the manuscript. Please see below for our point-by-point response to this input (blue) and manuscript revisions (red):

Reviewer #1 (Remarks to the Author):

In this report, the authors describe a novel method to isolate single platelets: droplet microfluidics. This method enables the characterization of single platelets and the investigation of individual platelet responses upon activation. The authors show that hyperreactive platelets within the total platelet population enhance the collective platelet response by paracrine signaling, thereby lowering the agonist concentration needed for maximal activation.

The authors present a methodology that is relevant for the further characterization of the individual platelet response. Droplet microfluidics can be a useful and relevant tool in future research towards platelet heterogeneity and platelet populations. I do however have some questions for the authors.

1. When looking at the vehicle controls of the single platelets, there appears to be a substantial number of platelets that have a relatively high fluorescence signal for PAC-1 and P-selectin labeling (Figure 3A-B, Suppl Fig 2 and Suppl Fig 5). This fraction is smaller in the collective response. The authors state that platelet activation was absent in the vehicle control, but can the authors exclude pre-activation/low-grade activation and desensitization of the platelets due to droplet generation. It would be helpful when the authors can include data on platelet activation of platelets that have been passed through the droplet microfluidics device without actual droplet generation to show that droplet generation does not affect platelet reactivity.

The reviewer raises a good point. These graphs were chosen to illustrate the complete dose-response trend between collectives and single platelets. The lengthy dose response experiments (for a limited post-draw experimental window (<4 hours)) and extra handling steps required for droplets did occasionally create elevated vehicle control signals. To provide confidence in our assertion we have added a new Supplementary Figure (1) with an example of equivalent droplet and bulk vehicle controls for both PAC-1 and P-selectin end-points. To ensure transparency across the principle data sets produced during this study we have accompanied this with a relative risk figure (a relative risk of 1 identifies similarity). Reviewing all experiments 5 of 12 PAC-1 droplet vehicle controls and 4 of 12 P-selectin were below 1 (but >0.8) indicating the additional steps required for droplet microfluidics can, on occasion, cause minor activation.

To address this critical issue we have expanded the text;

“Critically, platelet activation was minimal or absent in the vehicle control samples (Supplementary Figure 1) demonstrating that the shear conditions for droplet generation, and droplet transport⁴⁷, as well as the surfactant and fluorinated PDMS channel walls do not activate platelets.”

and added a cautionary note to the legend of Supplementary Figure 1:

“Supplementary Figure 1. Vehicle control comparison. The vehicle control signal from the droplet-confined platelets produces equivalent PAC-1 and P-selectin signal intensity distributions to platelet collectives (A). The relative risk scores comparing vehicle control single with collective platelet responses for the principle experiments involved in this study (B). **A score of 1 indicates no difference. A relative risk below 1.0 indicate minor activation, highlighting the need for careful handling during the additional steps required for droplet encapsulation.**”

We appreciate the concept of dissecting the fluidic components. Preliminary studies on transport and contact with microchannel materials (fluorinated PDMS coatings) did not result in platelet adhesion and aggregation. Either more in-depth investigations to identify causative components of droplet microfluidics are required or overall greater care is required with the additional steps required for droplet encapsulation: Importantly, the larger fraction (~60%) of droplet controls compared to collective vehicle controls had a relative risk score very close 1. The lower scores indicating (minor) activation we therefore consider to be the result of the extra handling steps, that with improved care (and ideally assay automation) can be avoided altogether.

Further, the description of the preparation of the collective samples (platelet count, dilution) and the protocol used to assess platelet activation in the collective samples, are unclear.

We have now provided text in the Materials and Methods to clarify the preparation and treatment methodology for the collective samples:

“The collective experiments were undertaken in microcentrifuge tubes and involved matched conditions to the droplet experiments, in which 25×10^6 /mL platelet samples were diluted 5-fold by the addition of equal volumes of agonist and antibody reagents and incubated for 15 minutes before fixation.”

2. The authors used the generation of larger droplets containing multiple platelets to show that PAC-1 and anti-P-selectin labeling are a good representation of platelet activation. They observe that platelets within the larger droplets -called minimal collectives- aggregate at moderate and maximal concentrations of convulxin (Figure 2H). However, when aggregation occurs, flow cytometry will not provide an accurate estimation of the platelet activation state (Figure 2I). Can the authors please comment?

To include aggregation in the flow cytometry read-out doublet gating was not applied allowing minimal collective and bulk collective dose response trends to emerge. PAC-1 staining (activated $\alpha_{IIb}\beta_3$) as an activation indicator, was not used as this has the potential to interfere with aggregation. We have clarified this in the Materials and Methods section:

“Here, doublet gating was not applied to retain platelet aggregates in the analysis and PAC-1 staining was not used to avoid interference with aggregation”.

The reviewer notes that the activation state becomes an average across the aggregate. Importantly, once aggregated they are in a fully active state (i.e. various activation states are no longer present).

Further, incubation with the antibodies was for 60 minutes. Why did the authors choose for 60 minutes incubation in the case of 50 μ m droplets/minimal collectives in comparison to 10-15 minutes incubation for the smaller droplets/single platelets?

The 60 minute incubation was chosen to allow aggregation to fully manifest. At the 15 minute time point aggregation was not evident in droplets by microscopy even with 100 ng/mL convulxin stimulation. Unlike, conventional light transmission aggregometry, stirring cannot be implemented during droplet incubation to accelerate the process. Furthermore, control experiments using autologous non-adjusted platelet rich plasma treated with the same final concentrations of convulxin in a plate reader (measuring light transmission) showed that at room temperature without stirring 60 min was optimal for measuring platelet aggregation. We have clarified this in the Materials and Methods section:

“In the absence of stirring, platelets were incubated for 60 minutes prior to emulsion breaking and fixation to allow aggregation to be concluded.”

3. The generation of single platelets using the droplet microfluidics device offers the opportunity to investigate the contribution of autocrine signaling to the individual platelet response. The authors also state that autocrine molecules inside the droplets enhance platelet activation. Have the authors looked at the effect of inhibition of the autocrine/paracrine TXA2 and ADP pathways in single platelets vs. platelet collective samples upon convulxin and TRAP stimulation?

The reviewer makes a good point involving the use of inhibitors to interpret autocrine/paracrine signaling effects. In preliminary experiments the TXA2 pathway was inactivated using in vitro aspirin (up to 265 μ M) treatment yet no appreciable differences were

observed in both the single and collective platelet conditions. This shows that more sophisticated experiments are required to accurately address this point and this is something we feel deserves a rigorous, in-depth study in its own right.

Concerning single platelets, placing them in 8-fold larger droplet volumes had minor apparent signal enhancement adding some weight to the self-feedback statement (i.e. greater dispersion leads to less autocrine feedback). Again this was preliminary research and we feel is not worth mentioning. Restricting ourselves to the presented data in the manuscript, we have added the following text sections to alert the readers to the speculation in our interpretation of this effect:

“This could indicate autocrine and paracrine signalling resulting from the accumulation of degranulation products within the droplets, but requires confirmation using inhibitors.” And;

“During confined platelet stimulation with convulxin degranulation results in the accumulation of stimulatory molecules in the droplets and this may lead to enhanced activation ($\alpha_{IIb}\beta_3$ activation) (Supplementary Figure 3). This observation deserves confirmation using inhibitors but nevertheless at the activation transition with a 3 ng/mL convulxin stimulation in droplets a clear bimodal distribution is evident with the activated population having a higher $\alpha_{IIb}\beta_3$ activation signal than platelet collectives also undergoing activation transition (0.3 ng/mL).” And;

“Elevated P-selectin signals in droplets relative to collective conditions are observed at 10 and 100 ng/mL convulxin. This could indicate autocrine and paracrine signalling resulting from the accumulation of degranulation products within the droplets, but again requires confirmation using inhibitors.”

4. The contribution of paracrine signaling to the overall platelet response becomes also apparent from the experiments where ADP is used as an agonist. The authors have chosen concentrations of ADP ranging from 0.01 to 10000 μ M. Why were such high concentrations of ADP (1000-10000 μ M) used? Can the authors explain the decrease in integrin activation and P-selectin expression?

The large dynamic range was initially chosen given the exploratory nature of using droplet microfluidics for the first time. Our reviews of the literature did not find examples of such extensive dose response experiments with ADP. Indeed, a large number use 1-4 lower concentrations as part of conventional light transmission aggregometry analyses. We cannot explain the desensitization but early work Baurand *et al* shows ADP-driven desensitization resulting from receptor internalisation. We have highlighted this issue to the readers using the following text and reference - added to the legend of Supplementary Figure 5:

Supplementary Figure 6. “Desensitization with 1 and 10 mM ADP stimulations may result from rapid receptor internalization¹.”

5. The authors show that dual stimulation of single platelets leads to the formation of an unresponsive third population of platelets (Figure 4A). However, stimulation of single platelets with the same concentration of convulxin (100 ng/mL) does result in full integrin activation after 15 minutes (Figure 2A). It would be interesting to see whether the third unresponsive population is only unresponsive towards annexin binding and was positive for PAC-1 binding at earlier timepoints. A time series experiment would clarify this.

We appreciate the reviewers point. From the outset the entire study was designed around using platelets that have not been washed, to better retain physiological functionality and kept consistent for extension from single to dual stimulation experiments. However, in doing so calcium needs to be applied to enable membrane inversion for annexin V binding, and therefore the further addition of rivaroxaban and GPRP to prevent coagulation. This changes the context of the experiment documented in Figure 4A from that of 2A so they are no longer comparable. It should also be noted that the integration of two activation pathways also alters sensitivity.

We have clarified the experimental context differences in the manuscript body:

“A dual stimulation dose response experiment was undertaken requiring the addition of calcium to unwashed platelets to enable membrane inversion for annexin V binding to phosphatidylserine moieties and also the addition of rivaroxaban to prevent additional thrombin formation by factor Xa and and gly-pro-arg-pro (GPRP) to limit the formation of fibrin fibres. The responses of single and collective platelet populations are compared using violin plots (Supplementary Figure 7) and show that the different conditions and integration of the two activation pathways alters system sensitivity (see Figure 2A): With platelet collectives pro-coagulant (annexin V high; $\alpha_{IIb}\beta_3$ low) and pro-aggregatory (annexin V low; $\alpha_{IIb}\beta_3$ high) heterotypic states emerged with a 100 ng/mL convulxin and 1.0 U/mL thrombin stimulation and is consistent with the literature.^{16,17,23,24”}

The reviewers comment to resolve, using earlier time points, the likely emergence of the PAC-1 signal followed by desensitization as the sub-population becomes polarized is excellent. However, unfortunately the current method is not suitable for short period measurements. Preliminary data (in collective systems) showed that the signal to noise ratio was substantially lower for incubations

smaller than 10 min making small changes in activation undetectable. Furthermore, the current state of the droplet method with encapsulation, collection, incubation and breakage with fixative makes incubation times smaller than 10 minutes unfeasible such that the proposed phenomenon cannot currently be investigated.

Minor comments:

Figure 2A-C: the range of the fluorescence intensity is different compared to the other figures in the manuscript and supplement (range: $10^1 - 10^3$ compared to $10^2 - 10^4$ in other figures).

Well spotted – we have gone back to cross reference the raw data, and recognize an error introduced during early efforts to import R data into AI. Now corrected to the $10^2 - 10^4$ scale.

Reviewer #2 (Remarks to the Author):

Jongen et al have applied microfluidic isolation of single platelets. As far as I know, this is a first time and an important step in trying to understand if, why and how the normal platelet population is composed of (functional) subpopulations.

Platelets are small but very abundant cells, roughly 5% of all cells in the human body are platelets (Sender et al, Cell 2016;164(3):337-40). Platelets are highly complex cells, on the morphological and functional level. During blood circulation, platelets serve homeostasis by passively and actively surveilling the vasculature. Integrity breaches are effectively sealed off by hemostasis, including an indispensable role for platelets. Platelets may further contribute to tissue healing as well, although that function is less understood.

Jongen et al have now used single cell isolation to investigate (possible) platelet heterogeneity, *in vitro*. In particular the authors have assessed the response of single platelets to increasing concentrations of known agonists. As an outcome measure the authors have used known reporter molecules expressed on the (activated) platelet surface using flow cytometry. The research question(s), the study design, the methodology and the written manuscript is strong. The data are very interesting and will advance the field and/or trigger new questions. The data furthermore lead to original conclusions. The data in Figure 2A-G in particular suggest that bulk platelet activation is “led” or “instigated” by a rare subpopulation of platelets that provides sensitivity gains to the bulk. These data stand out as the key experiment of the manuscript. I have a couple of remarks.

1. From the data in Figure 2A, I conclude that the ‘hypersensitive’ platelet subpopulation must be extremely rare. The reason for this conclusion is that the first resolution of this population in single cell analysis is technically only possible at 3 ng/mL of CVX. At this CVX concentration, bulk platelet preparations are fully activated. If these ‘hypersensitive’ platelets indeed sensitize the platelet bulk then they must have been present at 0.3 ng/mL as well (in bulk) while at that concentration not yet distinguishable in single cell analysis. At 0.3 ng/mL CVX the ‘hypersensitive’ platelets are thus either in the noise or just not present in quantities that allow resolution in flow cytometry, i.e. rarer than 1/100,000 assuming all events were read. If time proves the investigators right, then this concept is a landmark novelty. But the authors wisely do not claim that the ‘hypersensitive’ subpopulation is required for successful bulk activation even though their first sentence at L223 comes close. Therefore, this reviewer remains critical to the possible translation of these findings *in vivo*. One reason is that thrombocytopenia to low levels of normal (up to 5-8% or 20,000 platelets per μL) generally does not cause bleeding in otherwise healthy humans, meaning that the very rare subpopulations identified in this manuscript and seemingly important to sensitize bulk platelet activation would become even rarer (or almost non-existing) in these patients. Proving relevance of these findings to hemostasis/thrombosis is therefore challenging. The authors may want to briefly mention possible translational hypotheses in order to provide context and trigger the community in designing experiments that will address this.

We thank the reviewer for a thoughtful interpretation of our research. The so-called hyper-sensitive platelets are not as rare as 1/100,000 a feature resulting from 3-fold dose scaling and the broad-spectrum sensitivity variability. Instead, they are ‘hidden’ within the natural and experimental variation of the end-point signals. To highlight this we have used the following 2 sentences:

“These modes of paracrine signalling produce a spatiotemporal corraling effect that drives platelet cooperation to deliver the collective response. Nevertheless, sufficient numbers of activated platelets are required to polarise the entire platelet population into an activated response (e.g. Figure 2A; collectives with 0.1 ng/mL convulxin).”

The reviewer raises an important cautionary point about translation to *in vivo* contexts and we have added the following statement to the manuscript:

“While this holds promise for the identification of elements governing system behaviour and potential hubs for therapeutic intervention, caution is required for extension to *in vivo* contexts in which multiple cues integrated in time and space mediate responses that allow highly robust homeostatic control.”

2. The authors did well in designing a system that manages high throughput (10kHz) droplet formation. Yet, the first collected single

platelet still experiences a 5 minute longer dwell time (at least) than the final collected platelet. This 5 minute spread between the first and last platelet is unavoidable but nonetheless constitutes a study limitation. What is the chance that heterogeneity is introduced caused by this time gap? Do the authors agree that this is an experimental limitation? If not, please disprove and if yes, suggestion to mention this limitation in the manuscript.

The reviewer notes that absolute time matching is not possible with a continuous processing method versus a bulk processing method, potentially producing continuous variation in response, not a heterogeneous output as suggested. This temporal continuous variation is also overlaid by the natural continuous variability (see bulk data). However, within the chosen 10-15 minute incubation window variation was not apparent: In the early phases of the project this was investigated using a highly resolved time series. A window of 10-15 minutes produces equivalent signal intensities, and also produces equivalent and low noise levels that allow activation signals to be clearly discerned. See below for screen shots of this data. In summary, the detailed temporal analysis shows that this may introduce minor response broadening but not separation into heterotypic states.

Time-dependent PAC-1 signal emergence following stimulation with 100 ng/mL convulxin:

Figure A-15 Effect of incubation time of a pre-incubated with antibody platelet suspension in an agonist solution on signal to noise of stimulated and non-stimulated platelets. Platelet activity was measured with PAC-1 antibody and platelets were activated with 100 ng/mL of convulxin (red) compared to the vehicle control (blue). Platelets were incubated for **A) 0 min, B) 1 min, C) 2.5 min, D) 5 min, E) 7.5 min, F) 10 min, G) 12.5 min, H) 15 min, I) 17.5 min J) 20 min, K) 25 min and L) 30 min** before addition of fixative to stop the reaction.

Time-dependent P-selectin signal emergence following stimulation with 100 ng/mL convulxin:

Figure A-16 Effect of incubation time of a pre-incubated with antibody platelet suspension in an agonist solution on signal to noise of stimulated and non-stimulated platelets. Platelet activity was measured with P-selectin antibody and platelets were activated with 100 ng/mL of convulxin (red) compared to the vehicle control (blue). Platelets were incubated for **A)** 0 min, **B)** 1 min, **C)** 2.5 min, **D)** 5 min, **E)** 7.5 min, **F)** 10 min, **G)** 12.5 min, **H)** 15 min, **I)** 17.5 min **J)** 20 min, **K)** 25 min and **L)** 30 min before addition of fixative to stop the reaction.

We have added the following text:

“The 5 minute continuous droplet collection period followed by a 10 minute incubation produced overall droplet incubation times ranging from 10 to 15 minutes that were optimal for distinguishing activated platelets from the vehicle control platelets. Incubations can be extended to 60 minutes while retaining appreciable signal to noise cytometry data.”

3. At what platelet concentration were the ‘collective’ experiments performed? Was that also 25,000 per μL ? Please indicate the platelet concentration in the manuscript.

This has now been clarified in the M&M and in the comments to reviewer 1. In short, the same concentrations, dilutions and incubation periods were used for both droplet confinement and bulk (collective) experiments:

“The collective experiments were undertaken in microcentrifuge tubes and involved matched conditions to the droplet experiments, in which $25 \times 10^6/\text{mL}$ platelet samples were diluted 5-fold by the addition of equal volumes of agonist and antibody reagents and incubated for 15 minutes.”

4. Please mention the total number of (gated) events analyzed by flow cytometry in both conditions (single vs collective)?

We have now provided the number of analysed platelet per condition to the Materials and Methods section. The collective measurements were typically $\sim 48,000$ platelet events, while the droplet conditions varied, involving a range of numbers from 6,535 to 44,094 (average; $\sim 30,000$). We have added this to the Materials and Methods;

“For the single platelet experiments, an average of $\sim 30,000$ platelets events were measured for each condition and $\sim 48,000$ platelet events for each of the collective conditions.”

and the specific number ranges to each Figure legend:

“**Figure 2.** *Broad-spectrum response to convulxin stimulation and hypersensitive collective behaviour....* For each of the single and minimal collective conditions, 10,000–36,000 platelet events were measured, and $\sim 48,000$ for the collective conditions.”

“**Figure 3.** *Variable TRAP-14 response and increased collective sensitivity....* For each single platelet condition, 7,000–22,000 platelet events were measured, and $\sim 48,000$ for the collective conditions.”

“**Figure 4.** *Intrinsic heterotypic states in response to dual stimulation.....* For each single platelet condition, 14,000–26,000 platelet events were measured, and $\sim 48,000$ for the collective conditions.”

“**Supplementary Figure 5.** *Alpha and dense granule secretions correlate.....* For each single platelet condition, 22,000–44,000 platelet events were measured, and $\sim 47,000$ for the collective conditions.”

“**Supplementary Figure 6.** *Minor variability in the response to ADP and minor collective sensitivity gains.....* For each single platelet condition, 16,000–43,000 platelet events were measured, and $\sim 49,000$ for the collective conditions.”

“**Supplementary Figure 7.** *Dose responses for dual stimulated single and collective platelets.....* For each single platelet condition, 14,000–26,000 platelet events were measured, and $\sim 48,000$ for the collective conditions.”

5. For the ‘collective’ experiments using bulk activated platelets, were measures taken to avoid platelet aggregation (Tirofiban? Reopro)? Were platelet ‘aggregates’ found in these samples? What could possible artifacts be for the flow cytometry readout if (partial) aggregation was taking place?

The issue of aggregates and a consequence for cytometry is an important point. We did not use inhibitors to address this. Importantly, an increase in gated out events was not observed irrespective of the agonist concentration, a feature we attribute to the low platelet density used in the experiments and the short incubation time scales (10-15 minutes) without mixing. We have added the following text to the Materials and Methods section:

“Importantly, platelet dilution and short 15 minute incubations without stirring of the collective samples did not increase the gating out of platelet doublet/aggregate events, thereby surveying the population without bias.”

6. Please add “per mL” in Supplementary figure 1 below the syringes, in the first bullet, so to indicate platelet concentration instead of the unitless measure now.

Well spotted. Now amended.

7. L31 “a state of decay” is not what references 14,15 demonstrate. Maybe the authors intend to say that from the typical steady state of platelet bioproduction (IN) and clearance (OUT) automatically follows a heterogeneity in terms of senescence for any bulk platelet preparation isolated in a theoretical snapshot? Suggest to rephrase in order not use “decay”.

We appreciate that the language is somewhat deceptive, even with the special case of platelets and have accordingly changed this to the following:

“and, without a nucleus, having limited repair capabilities during ageing^{14,15} before clearance”.

8. L35 it's probably not necessary to refer to this many papers for COAT platelets (ref 22-24). Suggestion to retain one, preferably the milestone primary report.

We have now restricted the references to the seminal paper, Alberio *et al* 2000 paper.

9. L64 “platelet activation was absent in the vehicle control samples”. This seems correct from the data, but I had to look for it when reading this sentence. Suggestion to refer to a Figure here. A figure that contains these ‘validation’ data. Other reviewers/readers may not require this, but I found it confusing because the statement is important but seemingly without data when not referred to a figure. Only later on in the manuscript becomes clear that ‘untreated’ controls are included in every experiment.

Important point, now also addressed in response to reviewer 1, point 1.

10. L130-L141 is not a very strong paragraph in the paper, my remarks below may assist in making it stronger or have the authors decide not to keep it. In my opinion this paragraph does not add much to the overall finding.

We realise that a concentration error plus poor experimental definitions leads to query this paragraph, but we also find this a small, but important step to extending system interpretation from molecular alterations to aggregation, a higher order, functional response. Please see response in the subsequent points. As such we believe the findings and paragraph have merit and should be retained in a revised state. Discussed in response to subsequent points.

11. L133-135 This sentence is not supported by data “single aggregates of platelets are observed in droplets containing hypersensitive platelets, and not in droplets without hypersensitive platelets”. There is no panel in Figure 2 that supports this statement.

See response to point 12.

12. L134-L135 “At maximal concentrations (100ng/mL) all droplets contain platelet aggregates (Figure 2H)” needs more explanation or is poorly supported by the data. The labeling of droplets in panel H is green, white or unlabeled. Does this mean something? The outcome measure “aggregation” is not well described and therefore hard to interpret.

Building on points 10 and 11 and accommodating point 12 we now realise that certain steps of logic and experimental definitions are required to appreciate the significance of Figure 2(H). This builds on the previous data (Figure 2(B)) which shows, via the P-selectin end-point, the existence of some hypersensitive platelets when treated with 1 ng/mL convulxin. We have expanded the text to explain our reasoning:

“To confirm that the molecular $\alpha_{IIb}\beta_3$ activation and P-selectin end-points represent functional behaviour the dose response study was extended to larger droplets (65 μ L; ϕ 50 μ m) packaging 0–15 platelets. Here the existence of hypersensitive platelets was predicted to result in aggregation at moderate convulxin concentrations. At low concentrations (0.01 ng/mL) P-selectin negative platelets are observed as multiple, spatially distinct entities within each droplet. At moderate concentrations (1 ng/mL) this droplet case is also observed along with the other droplet case, each containing a single platelet aggregate. These stain positive for P-selectin, are typically large and, dictated by the Poisson statistic, must generally contain multiple, co-localised platelets (Figure 2(H)). Overall, this points to the existence of hypersensitive platelets in a large fraction of the droplets when treated with 1 ng/mL convulxin.”

We have also expanded the figure legend to clarify the labelling scheme:

“Triple fluorescent imaging (P-selectin, CD63 and CD42b) with brightfield overlay of minimal platelet collectives stimulated with 1 ng/mL convulxin. Droplets containing aggregates indicated by a green ring and those with multiple separate platelets by a white ring (H). Resulting dose response violin plots of minimal platelet collectives compared with bulk platelet collective responses (I).”

In addition, the figure caption mentions 1ng/mL and not 100ng/mL. Please clarify, suggestion to amend.

This is an error – signposting to Figure 2(H) should instead occur at the 1 ng/mL dose (as in the figure legend) and has now been changed. The order of the text is now changed to make it clearer which figure is being referred to.

13. L138-139 “This is indicative of autocrine signaling” is speculative at this instance. Suggestion to rephrase or to omit from this paragraph. The hypothesis of autocrine signaling is correctly mentioned in the final paragraph of the Results & Discussion section.

This has now been addressed, also in response to reviewer 1, where we clarify the speculation by adding the text “...This could indicate autocrine and paracrine signalling resulting from the accumulation of degranulation products within the droplets, but requires confirmation using inhibitors.”

14. L145-147 “Nevertheless, sufficient number of activated platelets are required to polarize the entire platelet population into an activated response (e.g. Figure 1A).” Do the authors mean Figure 2A? Figure 1A is a model of the study design. Please check.

Well spotted. Thank you. Now corrected.

15. L147 “our experiment involved platelets diluted to approximately 1/100 of in vivo concentrations.” Referral to my earlier remark (3). The platelet concentration reported in the methods section is 25×10^6 per mL. This is 25,000 per μL or 1/10 of normal, not 1/100. The concentration of one platelet per 8 pL is in fact three times higher than this, on average an equivalent of 125,000 per μL or 1/2 of normal. This needs to be addressed by the authors to avoid confusion and to provide relevance and context because the phrase that follows “suggesting digital activation may well occur” may no longer hold true.

We have more accurately described this as 1/50th of *in vivo* concentrations ($25 \times 10^6/\text{mL}$ diluted by agonist and antibody volumes to $5 \times 10^6/\text{mL}$ compared with $250 \times 10^6/\text{mL}$ in vivo). This is explained in lines 80-82. The one platelet per 8 pL indeed corresponds to $125 \times 10^6/\text{mL}$ (if all droplets occupied) and results from the Poisson statistic relevant to random encapsulation events. In this case platelets diluted to $5 \times 10^6/\text{mL}$ leads to approximately 4% of droplets containing one or more platelets.

16. L150 The authors should provide a strong reference that demonstrates that CD63 is a reliable reporter for dense granule secretion. Preferably not a review paper, but an actual original research paper that proves this statement. The CD63 receptor is expressed on platelet granule membranes and its density increases upon sustained platelet activation, but most papers mention this receptor as part of dense and lysosomal granules.

The reviewer raises a good point about labelling specificity; this also reports the lysosome fusion with the membrane. We have expanded the text and added a reference for this:

“Platelet cooperation is mediated through the secretion of alpha granules as evidenced by P-selectin exposure, but also ADP and serotonin secretion from dense granules (Supplementary Figure 5) as evidenced by CD63 presentation, a marker for dense granule and lysosome fusion with the membrane [REF: Nishibori, J. Clin. Invest 1993 91, 1775-1782]”

17. L151 “dense granule secretion pathway has a higher activation threshold” seems not to corroborate with reports on dense and alpha granule secretion kinetics in platelets. For instance, Jonnalagadda, D et al Blood 2012;120(26):5209-16 shows that serotonin (dense granule component) is released before PF4 (alpha granule component).

The reviewer raises a good point and although closely related we cannot directly reconcile sensitivity with kinetics (following activation). To alert the audience to this issue we have reworded the text as follows:

“The dense granule secretion pathway had a higher activation concentration than the alpha granule secretion pathway, although upon activation the kinetics of dense granule secretion are faster [Jonnalagadda, Blood, 120(26), 5209] which may allow the timely augmentation of pathways for specialized platelet activation.⁵¹⁻⁵³”

18. L153 “Again, autocrine signaling...” same remark as above (13). It is speculative at this instance. Suggestion to rephrase or to omit. The hypothesis of autocrine signaling is correctly mentioned in the final paragraph of the Results & Discussion section.

We think there may have been some confusion; the statement refers to the then Supplementary Figure 4 (not the then SI Fig 3). To avoid this confusion we have pushed mention of this earlier in the text, as a concise paragraph, and now becoming Supplementary Figure 3. In addition we have highlighted the potential merits of observing autocrine effects while also providing a cautionary statement that this is an artefact of the method when considering other biological questions:

“During confined platelet stimulation with convulxin degranulation results in the accumulation of stimulatory molecules in the droplets and this may lead to enhanced activation ($\alpha_{\text{IIb}}\beta_3$ activation) (Supplementary Figure 3). This observation deserves confirmation using autocrine inhibitors but nevertheless at the activation transition with a 3 ng/mL convulxin stimulation in droplets a clear bimodal distribution is evident with the activated population having a higher $\alpha_{\text{IIb}}\beta_3$ activation signal than platelet collectives also undergoing activation transition (0.3 ng/mL).”

19. L165 syntaxis “are”, should be “is”

Well spotted. Thank you.

20. L174-L176 “this scaling correlates ...”. The authors make a link with a possible in vivo time sequence for hemostasis here. But please provide an original research paper that proves this scaling in vivo. The reason for my skepticism is that during in vivo hemostasis, these separate steps are probably not that strictly time resolved as we tend to think from our in vitro models. Thrombin may be formed early by the tissue factor pathway in the gushing wound prior to significant platelet accumulation onto collagen. Amplification signals (ADP and/or Thromboxane A2) may be secreted from platelets following initial GPIb-VWF interactions (Cannobio I et al Cell Signal 2004;16:1329-44 and Liu J et al Blood 2005;106:2750-56) prior to firm arrest onto collagen. The time resolution described here appears very “textbook”, but may be too speculative in this context.

We appreciate the skepticism relating to the convenience of a textbook time line, that is not entirely consistent with the published body of work, and choose to remove this sentence.

21. L191-L205, Figure 4 and L305-L309 mention the use of GPRP as an inhibitor for coagulation. I understand that some way of inhibiting coagulation is required in this experiment using Ca²⁺. However, GPRP may also influence PS exposure on platelets and influence the data shown in Figure 4 and Figure S6. Check Brzoska et al, PLoS ONE 2013;8(2):e55466 (DOI 10.1371/journal.pone.0055466). Did the authors try using only DOACs (like rivaroxaban or edoxaban or alike)? I don't consider this a major issue, but GPRP is not very specific and may stick or bind causing artifacts to normal platelet activation.

Coagulation was indeed prevented using 0.5 uM rivaroxaban and 100 mM GPRP, and found that both were required to prevent activation of the vehicle control. However, this treatment was used for both droplet confined and collective platelet conditions such that the central, different response findings cannot be attributed to this.

22. L223-236 is a strong and well balanced paragraph, this is a good way to interpret the data.

Thank you.

Reviewer #3 (Remarks to the Author):

The manuscript by Jongen and colleagues describes the use a droplet microfluidic platform to isolate single platelets to investigate the differences between the single cell and collective response of platelets under different forms of stimuli. The authors devised a clever approach using droplet microfluidics to isolate the platelets to prevent paracrine signaling that occurs in a bulk suspension (herein referred to as a collective). This approach allows for the interrogation of single cells in the absence of paracrine signaling. After isolate, the platelets were fixed, stained, and screened using flow cytometry. The authors found that the hypersensitivity of the collective population of platelets was driven by a small hypersensitive subpopulation of platelets by comparing the responses of single and collective cells monitored by integrin and P-selectin activation. The authors postulate that this sensitivity is a function of the concentration and choice of agonist used to stimulate the platelets. Overall, the manuscript is well-written and easy to follow with clear and easy to interpret figures. This reviewer has a few minor comments that the authors should address prior to publication.

1) At the end of the manuscript the authors mention potential temporal control of the system and that the exposure time to an agonist and the length of time a platelet is isolated from the collective could further influence the activation of the selected biomarkers. In the manuscript, the authors only selected a single time point. Can they provide some insight on why a 10 min incubation time was selected and what they would expect to see if the droplet incubation time was increased or decreased?

In the early phases a preliminary time series experiment was used to identify an optimal 10-15 minutes window that produced equivalent signal intensities, and also produces equivalent and low noise levels that allow activation signals to be clearly discerned. See response to reviewer 2.

“The 5 minute continuous droplet collection period followed by a 10 minute incubation produced overall droplet incubation times ranging from 10 to 15 minutes that were optimal for distinguishing activated platelets from the vehicle control platelets. Incubations can be extended while retaining appreciable signal to noise cytometry data.”

2) Related to the above comment, can the authors provide insight on platelet viability in the droplets for the 10 min incubation and potentially greater incubation times? What is the upper threshold of the microfluidic system?

The reviewer raises an important point that would extend the method to other hypotheses. Platelets have been incubated for 30 and 60 minute time scales in droplets without desensitization or functional losses as measured by cytometry and also by microscopy (aggregation). We have added a short statement to explain the upper 60 minute time scale.

“In the absence of stirring, platelets were incubated for 60 minutes prior to emulsion breaking and fixation to allow aggregation to be concluded.”

Returning to the central question, these experiments did not involve direct viability reporters such that time-dependent viability remains speculative. Nevertheless, it is well known from the droplet literature that mammalian cells can be encapsulated in a viable state within droplets for lengthy periods (>24 hours) based on the availability of sufficient volumes to provide nutrients and accommodate waste products.

3) Can the authors provide insight on how the droplet breaking procedure affected platelet viability and platelet recover? Do they have any insight on platelet losses and/or changes in platelet response/activation using the system?

This is a very interesting consideration that has the potential to allow stochastic platelet activation behavior to be investigated. The breaking procedure removes surfactant and oil containment, immediately transferring all platelets into a large pool of fixative. Platelets were retrieved in abundant numbers for cytometry indicating minimal losses. The platelets could be broken into other aqueous environments allowing viable platelets to be recovered. However, we are tentative with claims concerning viability or retaining functionality without direct empirical evidence.

To clarify the concern regarding platelet losses during breaking the emulsion:

“...then combined with CellFix fixative (BD Biosciences) and subsequently with 1H,1H,2H,2H-perfluoro-1-octanol (PFO, Sigma-Aldrich) to destabilise the droplet interface and break the emulsion with minimal platelet losses during extraction of the aqueous volume containing fixed platelets.”

4) Can the authors report the total cell numbers analyzed for each of the experiments? It would be helpful to see how many cells were analyzed for the single platelet versus the collective experiments.

We have now provided the number of analysed platelet per condition to the Materials and Methods section. The collective measurements were typically ~48,000 platelet events, while the droplet conditions varied, involving a range of numbers from 6,535 to 44,094 (average; ~30,000). We have added this to the Materials and Methods;

“For the single platelet experiments, an average of ~30,000 platelets events were measured for each condition and ~48,000 platelet events for each of the collective conditions.”

and the specific number ranges to each Figure legend:

“**Figure 2.** *Broad-spectrum response to convulxin stimulation and hypersensitive collective behaviour....* For each of the single and minimal collective conditions, 10,000–36,000 platelet events were measured, and ~48,000 for the collective conditions.”

“**Figure 3.** *Variable TRAP-14 response and increased collective sensitivity....* For each single platelet condition, 7,000–22,000 platelet events were measured, and ~48,000 for the collective conditions.”

“**Figure 4.** *Intrinsic heterotypic states in response to dual stimulation.....* For each single platelet condition, 14,000–26,000 platelet events were measured, and ~48,000 for the collective conditions.”

“**Supplementary Figure 5.** *Alpha and dense granule secretions correlate.....* For each single platelet condition, 22,000–44,000 platelet events were measured, and ~47,000 for the collective conditions.”

“**Supplementary Figure 6.** *Minor variability in the response to ADP and minor collective sensitivity gains.....* For each single platelet condition, 16,000–43,000 platelet events were measured, and ~49,000 for the collective conditions.”

“**Supplementary Figure 7.** *Dose responses for dual stimulated single and collective platelets.....* For each single platelet condition, 14,000–26,000 platelet events were measured, and ~48,000 for the collective conditions.”

5) This reviewer had a difficult time seeing the difference between the single and minimal collective populations in figure 2. Another color for the minimal collective data would make it easier to see the difference.

Good suggestion – the colour has been changed to pink to ally the data sets to the red collective condition, yet underscore the minimalistic nature.

REVIEWERS' COMMENTS:

Reviewer #1 (Remarks to the Author):

The authors addressed most of my comments. I only have a few minor concerns that deserve attention.

1. Regarding point 2, I am not fully convinced of the authors' response to my comment stating that flow cytometry will not provide an accurate estimation of the platelet activation state when aggregation occurs. The authors replied that once platelets aggregated, they are in a fully active state (i.e. various activation states are no longer present). When looking at Figure 2I, I however notice a large spread in P-selectin staining, especially at a convulxin concentration of 1.0 ng/mL, suggesting that the platelets are not equally active. Moreover, how do the authors reconcile their statement regarding the fully active state of all platelets in an aggregate with their concept of platelet heterogeneity where hypersensitive platelets provide sensitivity gains to the overall platelet population and platelet population polarization.

2. The authors included a cautionary statement concerning the minimal activation of the vehicle control due to droplet generation in the text and supplements. Here, I would like to mention Figure A-15 included in the response to reviewer 2. If I understand correctly, this is a time series of PAC-1 binding in response to convulxin stimulation in platelet collectives. There appears to be substantial integrin activation in the vehicle control over time. Can the authors please comment?

Reviewer #2 (Remarks to the Author):

No more comments from my side. Great work, congratulations to the authors.

Reviewer #3 (Remarks to the Author):

This reviewer thanks the authors for their efforts to address the concerns from the initial review. This reviewer feels the manuscript is ready to be accepted.

Response to Reviewers Feedback:
manuscript ID: COMMSBIO-20-0383

4th May 2020

Dear Reviewer#1,

Thank you once again for your diligence and support in improving our manuscript. Please find below our response to your 2 outstanding concerns:

1. Regarding point 2, I am not fully convinced of the authors' response to my comment stating that flow cytometry will not provide an accurate estimation of the platelet activation state when aggregation occurs. The authors replied that once platelets aggregated, they are in a fully active state (i.e. various activation states are no longer present). When looking at Figure 2I, I however notice a large spread in P-selectin staining, especially at a convulxin concentration of 1.0 ng/mL, suggesting that the platelets are not equally active. Moreover, how do the authors reconcile their statement regarding the fully active state of all platelets in an aggregate with their concept of platelet heterogeneity where hypersensitive platelets provide sensitivity gains to the overall platelet population and platelet population polarization.

We agree that in the minimal collective versus collective convulxin dose response experiments with P-selectin measures results in a large signal spread for both conditions. We appreciate that this is visually misleading in Figure 2i without qualifying text. In these experiments, a 60 minute incubation period was used (not 15 minutes) to allow aggregation to conclude and doublet(+) gating was removed to include aggregates in the cytometry read-out. The data spread, notably substantial increased intensity, results from either single inactive platelets, single active platelets and then various aggregate scales (i.e. 2 to 15 platelets in the minimal collective) containing platelets that are presumed to be fully active but will each have different signal intensities that are combined as an aggregate into an ensembled data point. To qualify this, inactive single platelets (dispersed, not aggregated) within the larger droplets do not stain for P-selectin and CD63 as evidenced by fluorescent microscopy. In comparison the platelet aggregates staining positive for P-selectin and CD63 were evidently of different sizes and thus different area-summed intensities that will be interpreted as single flow cytometry events with wide-ranging signal intensities. However, the resolution of the microscopy does not allow the possibility to discern different regions with different intensities (i.e. individual platelets with different signal intensities in the same aggregate). The following text has been added to the legend of Figure 2 address this issue:

" To measure aggregates by flow cytometry, doublet(+) gating was removed, thereby increasing the signal spread by the inclusion of various aggregate scales (2–15 in the case of minimal collectives) along with signals from active single platelets and inactivate single platelets."

2. The authors included a cautionary statement concerning the minimal activation of the vehicle control due to droplet generation in the text and supplements. Here, I would like to mention Figure A-15 included in the response to reviewer 2. If I understand correctly, this is a time series of PAC-1 binding in response to

convulxin stimulation in platelet collectives. There appears to be substantial integrin activation in the vehicle control over time. Can the authors please comment?

There is indeed substantial activation, extending from a maxima of 10^3 with 1 minute to 2×10^4 by 30 minutes. A similar pattern is observed with the P-selectin end-point. This is a frustration we encountered during assay development and we can only conclude is a consequence of prolonged *ex vivo* incubation in the context of antibodies and/or associated buffers and reagents. As such we chose to define an optimal 10-15 minute incubation period in the Methods section. With the longer, 60 minute incubations relevant to the minimal collective experiments to allow aggregates to assemble this issue is also problematic. To some extent, the inclusion of multiple platelets in each cytometry data point allows the signal to be elevated and appreciably discerned from the noise.

“The 5 minute continuous droplet collection period followed by a 10 minute incubation produced overall droplet incubation times ranging from 10 to 15 minutes that were optimal for distinguishing activated platelets from the vehicle control platelets. Incubations can be extended to 60 minutes while retaining appreciable signal to noise cytometry data.”

Reviewer #2 (Remarks to the Author):

No more comments from my side. Great work, congratulations to the authors.

Thank you for your time and compliment; we're very proud of the research.

Reviewer #3 (Remarks to the Author):

This reviewer thanks the authors for their efforts to address the concerns from the initial review. This reviewer feels the manuscript is ready to be accepted.

Thank you for your time and endorsement of our manuscript.

We hope these inclusions and clarifications satisfy the concerns raised by reviewer#1 to make for an improved and finalised manuscript in readiness for formatting.